# Healthcare-Seeking Behavior among Chinese Older Adults: Patterns and Predictive Factors

**DOI:** 10.3390/ijerph18062969

**Published:** 2021-03-14

**Authors:** Yanbing Zeng, Yuanyuan Wan, Zhipeng Yuan, Ya Fang

**Affiliations:** State Key Laboratory of Molecular Vaccinology and Molecular Diagnostics, School of Public Health, Xiamen University, Xiamen 361102, China; ybingzeng@163.com (Y.Z.); wyymua@126.com (Y.W.); swjtueryzp@163.com (Z.Y.)

**Keywords:** health services use, socioeconomic status, self-rated health, population aging, healthcare-seeking behavior

## Abstract

This study aimed to investigate the patterns and predictive factors of healthcare-seeking behavior among older Chinese adults. A sample of 10,914 participants aged ≥60 years from the 2011, 2013 and 2015 China Health and Retirement Longitudinal Study (CHARLS) was included. The bivariate analyses and Heckman selection model was used to identify predictors of healthcare-seeking behavior. Results shows that the utilization rate of outpatient services increased from 21.61% in 2011 to 32.41% in 2015, and that of inpatient services increased from 12.44% to 17.68%. In 2015, 71.93% and 92.18% chose public medical institutions for outpatient and inpatient services, 57.63% and 17.00% chose primary medical institutions. The individuals who were female, were younger, lived in urban, central or western regions, had medical insurance, had poor self-rated health and exhibited activity of daily living (ADL) impairment were more inclined to outpatient and inpatient services. Transportation, medical expenses, the out-of-pocket ratio and the urgency of the disease were associated with provider selection. The universal medical insurance schemes improved health service utilization for the elderly population but had little impact on the choice of medical institutions. The older adults preferred public institutions to private institutions, preferred primary institutions for outpatient care, and higher-level hospitals for hospitalization.

## 1. Introduction

Healthcare-seeking behavior, or illness-related behavior, has been defined as the actions taken by individuals who perceive they have an illness to obtain a suitable remedy [1]. This behavior involves a series of decision-making processes governed by both individual characteristics, beliefs and provider-related features. During this process, decisions on whether to seek treatment, from whom to seek treatment, what kind of treatment to seek, as well as how many healthcare resources to use are usually made. Thus, in theory, an individual’s healthcare needs do not necessarily turn into effective demand. Understanding people’s healthcare-seeking behavior is crucial for improving universal health coverage, as it also concerns the efficiency of the national healthcare system. A wide range of policy makers, hospital administrators, and insurers all need to understand the patterns and driving forces behind patients’ behaviors to design healthcare policies or programmes [2].

### Ageing and Healthcare-Seeking Behavior in China

As a result of the increased life expectancy and declining fertility rate, China has experienced an unprecedented increase in the rate of population ageing in recent decades. Since entering the aging society at the end of the 20th century, China has become the country with the largest elderly population and the fastest growth rate [3]. In 1999, the percentage of people aged 60 years or older was only approximately 10% in China, which was approximately the same as the worldwide average in 1999 and well below that in most developed countries. By 2018, the number of people aged over 60 almost doubled from 125 million to 250 million, accounting for 17.9% of the total population. It took many more years for this magnitude of increase to occur in the United States, Japan, and many other countries. In the near future, as the baby boomers who were born in the second half of the last century reach old age, the rapid ageing process will continue to increase until 2050.

Because increasing age is expected to be associated with an increasing need for healthcare, one immediate effect of China’s rapid aging process might be increased healthcare utilization. One glaring issue is that the elderly is at a risk of catastrophic illness, such as cancer, heart attack or stroke [4,5,6], imparting a medical, emotional and fiscal burden on society [5,6]. The fifth national health services survey shows that the prevalence of two-week long illnesses and chronic conditions among Chinese people aged 60 and over are 56.9% and 71.8%, respectively, which are much higher than that of other age groups and are associated with an accelerating trend. Nearly seven-tenths older people suffer from hypertension, diabetes, cerebrovascular disease, ischemic heart disease, or chronic obstructive pulmonary disease. Additionally, 16.2% of elderly people experience more than one chronic condition concurrently. Moreover, although the life expectancy is increasing, older people are not healthy during the additional years of life. In 2018, the Chinese life expectancy was 77 years, and the healthy life expectancy was 68.7 years, which means that older people spend 8 to 9 years on average living with diseases. Therefore, the older population is becoming the largest consumer group of healthcare services among all age groups. This is putting additional pressure on the existing healthcare arrangements in China.

In the early years, China’s healthcare system was provided by hospitals, especially public hospitals, with 90% of the total number of in-patient and out-patient medical services [7]. After several rounds of reform were implemented, China achieved large improvements in access to healthcare services, and the private healthcare sector has continued to expand in China [8,9]. However, unexpectedly, although private facilities have been estimated to constitute 46.7% of all healthcare facilities nationwide, the share of health services provided by private providers was determined to have only increased to approximately 10% in 2013 [10]. The distribution of medical resources in urban and rural areas in China is uneven, and the urban health investment, medical institutions and employees are far more than those in the rural areas, the medical treatment probability of the rural elderly is significantly lower than that of the urban [11], and a large number of elderly people still cannot afford formal treatment due to the high medical costs, especially older people who are vulnerable regarding financial safety. In the decade prior to 2013, although the rate of patients who were in need of hospitalization but were not hospitalized decreased, 20% of the Chinese elderly people remained without hospitalization. Unmet healthcare needs have become an increasingly large threat to population health.

In general, Chinese healthcare policy was intended to provide structure for a hierarchical medical system but not involve a strict general practitioner design [12]. In other words, patients are free to choose the kind of healthcare provider they wish to receive treatment from. Although the current universal medical insurance schemes follow a differential reimbursement policy, leading the same treatments provided in high-level facilities to have low reimbursement rates, many patients tend to choose large hospitals and well-known doctors, regardless of their disease severity. It is common for large tertiary hospitals to be overcrowded while primary healthcare facilities are under-utilized, which results in waste and inefficient use of medical resources. Such waste also occurs in the private healthcare sector. Since the healthcare reform was initiated in 2009, both the number and scale of private healthcare providers have been close to those of public hospitals. Nevertheless, the burgeoning number non-public healthcare providers have failed to yield the same level of increase in service provision, which account for only 10% of the Chinese healthcare market [13]. In principle, patients in China can walk in and obtain treatment at any level of healthcare facility of any ownership without restrictions, especially in the case of outpatient care services [8]. The reason why private health services are not growing as fast as private healthcare facilities remains unknown. This has led to increasing interest in patient choice between private and public providers in China [14]. An ample number of studies have been carried out in China to investigate old people’s healthcare utilization regarding many aspects; however, little attention has been given to their choice of healthcare providers. Previous studies have discussed the relationship between medical insurance types and healthcare utilization [15,16,17] or medical expenditure [18,19,20]. Other studies have focused on the inequalities in health service utilization by the patient’s sex, residence, or socioeconomic status [11,21,22]. Nevertheless, only a few domestic studies have examined on what basis senior patients choose their healthcare providers. The very first work by Yip, Wang, and Liu (1998) identified income, insurance and health perception as co-determinants of provider choice in the previous three-tier healthcare system but was limited to patients in a single hospital [23]. Brown and Theoharides (2009) analyzed data from 25 counties using a nested logit model and found that hospital choice was influenced by the reimbursement scheme and household income [24]. Another study carried out a discrete choice experiment to examine the effect of disease severity, time and medical cost, providers’ equipment and skills on rural residents’ preferences for healthcare facilities [25]. The scope of these studies was limited to a particular facility and otherwise failed to provide empirical evidence using national representative data after the latest reform was implemented in 2009. However, these studies have noted some significant factors that can help predict hospital choices. A more recent study investigated the impact of health insurance and socioeconomic status on the choice between public and non-public healthcare providers among patients aged over 45 by using data from a nationwide survey [13].

The Andersen’s behavioral model of health service use is the most classic model in the field of healthcare service forecasting internationally, we use it as an interpretive framework to investigate the patterns and factors explaining healthcare-seeking behavior among Chinese elderly people. The direction and magnitude of the predisposing characteristics, enabling resources, and healthcare needs in the model were verified, and the conclusions are internationally comparable. It also verified its applicability to the study of the elderly’s medical treatment behavior in China, and made a useful transformation and exploration for the extensive use of this model in China. We focused on two categories of behaviors: healthcare utilization and provider choices. We used both outpatient and inpatient services in the analysis. Specifically, we compared old people’s healthcare-seeking behavior in spatial and time dimensions. Heckman’s sample selection model was applied in accordance with previous research to address potential selection bias [13,26].

## 2. Materials and Methods

### 2.1. Data

This study used data collected in 2011, 2013 and 2015 of China Health and Retirement Longitudinal Study (CHARLS). Patterned on the Health and Retirement Study conducted in the US and its sister surveys conducted in other countries, CHARLS is a nationally representative panel survey of adults over 45 and their spouses living in households. The baseline CHARLS survey was administered in 2011 and included over 17,500 individuals in 150 counties/districts and 450 villages/resident communities in 28 of 34 Chinese provinces. A four-stage, stratified, probability proportional to size sampling technique was adopted to obtain the sample. The baseline cohort was followed up every two years (2013, 2015) using a face-to-face computer-assisted personal interview. Additional detailed descriptions of the cohort have been published [27].

The 2015 wave of CHARLS was a second follow-up survey and involved 21,789 respondents in total. Since we were interested in the healthcare-seeking behavior of elderly people, we restricted our analysis to the subsample comprised of 10,914 respondents aged 60 and above. After excluding the individuals with key variables missing or illogical answers, 10,172 respondents were finally included in this study.

We use the data of 2011, 2013 and 2015 to analyze the change trend of medical-seeking behavior of the elderly in China, and further analyze the influencing factors of medical seeking behavior of the elderly in 2015.

### 2.2. Andersen’s Behavioral Model of Health Service Use

On the basis of the Anderson behavior model, this paper constructed the adjusted Andersen theoretical model of influencing factors on healthcare-seeking behavior according to the research purpose and variable information availability (Figure 1). The predisposing characteristics were gender, age, region, urban residence, educational background, marital status, drinking habits, and smoking habits. The enabling resources consisted of income, medical insurance, pension and intergenerational support. The need factors were self-rated health status, chronic condition, and functional limitations.

### 2.3. Measures

This study had two outcome measures: reported healthcare utilization and reported choice of healthcare settings or providers. Reporting healthcare utilization is measured by (1) visiting a doctor for outpatient care in the last month; (2) receiving inpatient care in the past year. Choice of healthcare providers is measured by (1) whether the healthcare provider the respondent visited most recently for outpatient care during the last month/or for hospitalization in the past year was public or private, among which, public hospitals refer to the hospitals organized by the government and included in financial budget management; private hospitals refer to non-governmental public hospitals of a private nature. In China, most of the private hospitals are health institutions funded by the society and run by profit-making organizations; there are also a few non-profit organizations that enjoy government subsidies.; (2) whether the healthcare provider above was a primary healthcare facility, which comprises community healthcare centers, township hospitals, healthcare posts and village clinics. All outcome variables are coded as 1 if yes, 0 otherwise. A modified Andersen behavioral model of health service use was employed as a theoretical framework to examine factors associated with healthcare-seeking behavior. It proposes an explanation for healthcare services utilization based on individuals’ predisposing characteristics, enabling resources, and healthcare needs [28]. This model has been widely used in studies exploring the determinants of healthcare utilization, hospital choice, medical costs, cancer screening and others [13,29,30,31].

The predisposing characteristics were gender, age, region, urban residence, educational background, marital status, drinking habits, and smoking habits. We also included the respondents’ attitudes towards the quality, cost, and convenience of local healthcare services, which were rated on a 5-point Likert-type scale (from 1 = very satisfied to 5 = very dissatisfied). The enabling resources consisted of income, medical insurance, pension and intergenerational support. Medical insurance is measured by having any type of medical insurance (0 = no insurance, 1 = UEMI (Urban Employee Medical Insurance), 2 = URMI (Urban Resident Medical Insurance), 3 = NCMS (New Rural Cooperative Medical Scheme), 4 = other types of insurance). The need factors were self-rated health status, chronic condition, and functional limitations. Functional limitations are assessed with activities of daily living (ADLs) and instrumental activities of daily living (IADLs). ADLs involve six basic tasks: dressing, bathing or showering, eating, getting into or out of bed, toileting, and controlling urination and defecation. IADLs are composed of five higher-ordered daily tasks: doing household chores, preparing hot meals, grocery shopping, managing money, taking medications. Having difficulty with any ADL or IADL because of health and memory problems is coded as 1, 0 otherwise. We also incorporated transportation availability, medical expenses and other factors occurring with outpatient services or hospitalization.

### 2.4. Analysis

The sample selection often arises from practice because of the partial observability of the outcome variable. When sample selection is present, the observed data do not represent a random sample of the population, even after checking explanatory variables, that is, the data missing in the sample does not respond to a random selection process [32]. Thus, a standard analysis using only complete cases will lead to biased results. In order to overcome the possible sample selection bias and obtain the unbiased estimation of medical treatment behavior, we use the sample selection model proposed by James Joseph Heckman. The Heckman model was applied to account for possible selection issues, expressed as two stages: (1) the selection stage, in which a probit equation is used to predict the likelihood that an older adult will use healthcare services and calculate an inverse Mills ratio for each observation, indicating the instantaneous probability of an observation being selected for the sample [33,34], and (2) the outcome stage, in which the choice of healthcare settings or providers, which is conditional on the respondents’ reported healthcare utilization, is estimated. Assuming that the error terms of two equations are independent of each other, this stage includes the inverse Mills ratio as a predictor to check and correct for bias caused by sample selection.

The two stages are written as:(1)$$S_{i}^{*}=\sum_{k}Z_{ik}\alpha_{k}+\alpha_{0}+\varepsilon_{i\;},\;\;S_{i}=\{\begin{array}{c} 1\;if\;S_{i}^{*}>0\\ 0\;if\;S_{i}^{*}\leq 0 \end{array}$$
(2)$$E\left(Y_{i}|S_{i}=1,Z\right)=\;\sum_{j}X_{ij}\beta_{j}+\beta_{\lambda }\lambda_{i}+\beta_{0}+\mu_{i}$$
where $S_{i}^{*}$ is a latent variable of underlying probability of reporting healthcare utilization, $Y_{i}$ is the elderly’s choice of healthcare providers. $Y_{i}$ is observable only when an old adult used healthcare services ($S_{i}=1$).$\;Z_{ik}$ and $X_{ij}$ are the vectors of factors predicting healthcare utilization decision and the choice of healthcare provider. $\lambda_{i}$ is the inverse Mills ratio. $\alpha_{0}$ and $\beta_{0}$ are constant terms, $\varepsilon_{i}$ and $\mu_{i}$ are error terms. The Heckman model allows for the existence of a correlation for the random perturbation terms of the two equations, and the regression coefficient for λ term introduced in Equation (2) is actually the correlation coefficient of the two, when used as a parameter estimate for “$\beta_{\lambda }=0$” of the null hypothesis to perform a likelihood ratio test. If the null hypothesis is rejected, the coefficient $\beta_{\lambda }$ is significantly different from 0, it is considered to be subject to sample selection bias and requires a correction using the Heckman model. If this is not considered significant, it is estimated directly using the general probit model.

All statistical analyses were performed using STATA MP-64 statistical software 15.0 (StataCorp. LP, College Station, TX, USA).

## 3. Results

### 3.1. Sample Characteristics

Sociodemographic and health-related statistics of the study sample are given in Table 1. Among the 10,172 Chinese elderly people investigated, 49.48% were male, and 10.43% were old. More than 70% of the respondents lived in rural areas, and over half had no formal education. Nearly 20% of respondents were divorced, widowed, or single. Approximately 10% had no medical insurance, and approximately 83% were enrolled in universal medical insurance schemes, while the remaining 6.57% had other types of insurance. Of the respondents, only 23.50% reported their health status as good, while 39.25% had difficulty with ADLs or IADLs.

### 3.2. Patterns of Healthcare-Seeking Behavior

Figure 2 presents the healthcare-seeking behavior among Chinese elderly people from 2011 to 2015. The proportions of senior patients receiving outpatient and inpatient services both increased steadily. The rate of use of outpatient services increased from 21.61% to 32.41%, and that of inpatient services increased from 12.44% to 17.68%.

Public providers were consistently the first choice for individuals seeking formal healthcare, accounting for at least 70% of the outpatient services provided and 90% of the hospital services provided. In particular, the proportion of elderly people who chose public medical institutions for outpatient care was 90.56% in 2011, 70.72% in 2013, and 71.93% in 2015. A large decrease occurred between 2011 and 2013. The proportion of elderly people who chose to be hospitalized in public institutions was 94.26%, 94.01% and 92.18% in the same years, respectively.

With regard to the different hospital levels, more than half of the elderly people received outpatient care in primary facilities, whereas over seven-tenths chose higher-level hospitals when in need of hospitalization. In 2011, 67.66% and 26.99% of senior patients received outpatient and inpatient care, respectively, in primary healthcare facilities. However, after a steady decline throughout the 4-year period, only 57.63% and 17.00% of the elderly people chose to receive outpatient and inpatient care, respectively, in primary healthcare facilities in 2015.

As shown in Table 1 and Table 2, 32.41% of the study population had visited a doctor for outpatient services in the last month, among whom 72.93% chose a public healthcare provider and 57.63% chose a primary healthcare facility such as a community healthcare centre, township hospital, healthcare post and village clinic. During the past year, 17.68% of the elderly people had received inpatient services. Of them, 92.18% were hospitalized in a public hospital, and 17.00% were in a primary healthcare facility. This suggests that almost all elderly people chose to use public inpatient services, while the hospitalization rate in high-level hospitals was approximately 6 times higher than that in primary facilities.

Compared to the elderly individuals who did not utilize healthcare services, those who did were more likely to be female, aged over 80, urban residents, unmarried, have universal medical insurance or another type of medical insurance, and have a poorer health status. The elderly people with UEMI and other types of insurance were more likely to have outpatient visits and inpatient care compared with those with URMI, NCMS and no insurance. When choosing health providers for outpatient services, the individuals who were female, oldest-old, and ADL impaired and those with a lower socioeconomic status relied more on private and primary healthcare facilities. In regard to the choice for hospitalization facilities, nearly all subsamples had the same preference for public hospitals. Moreover, the individuals who lived in rural areas, were less educated, had no insurance or were covered by NCMS were hospitalized more often in primary healthcare facilities.

To better understand the patterns of healthcare services utilization across areas, a geographic representation of the prevalence of outpatient and inpatient service use is presented in Figure 3. In 2015, elderly people who lived in central and western areas were more likely to utilize healthcare services than those in eastern areas (outpatient: 33.85% vs. 30.60%, inpatient: 18.70% vs. 15.22%). Among all 28 provincial areas, Beijing had the highest proportion of outpatient service use (53.2%), while Guizhou, which is located in western China, had the lowest (22.3%). The northwest region near the border, Xinjiang, was the province with the highest hospitalization rate (31.6%), while Beijing was the lowest (6.4%).

### 3.3. Predictors of Healthcare-Seeking Behavior

#### 3.3.1. Healthcare Services Utilization

The results of the multiple regression analysis regarding what factors affect outpatient and inpatient service utilization in elderly people are shown in Table 3. The coefficients indicate that the likelihood of utilizing outpatient services increased significantly with income, having medical insurance and pension, the number of children alive and their economic support, worse self-reported health, more chronic diseases, and ADL difficulty. Most of the predisposing characteristics had no significant association with outpatient use. The factors promoting inpatient utilization included being male, being aged over 80 years, income, having UEMI, the number of children alive, worse self-reported health, more chronic diseases, and ADL difficulty. Drinking and dissatisfaction towards local healthcare decreased the possibility of hospitalization.

#### 3.3.2. Choices of Healthcare Provider

In this part, factors determining older adults’ preference between public and private healthcare providers and between primary and high-level facilities were investigated. In addition to the independent variables contained in Andersen’s model, factors occurring with healthcare such as transportation availability and medical expenses were added in the analysis.

Taking visiting a public healthcare provider as the dependent variable, $\beta_{\lambda }$ estimated from the Heckman selection model was not significantly different from 0; therefore, a probit model was carried out with the individuals reporting healthcare utilization, as no sample selection bias was present. The results are shown in Table 4. Urban, married elderly people were more likely to visit a public outpatient care provider. Receiving outpatient services in public hospitals was not only related to emergency treatment and higher medical costs but also related to lower out-of-pocket portions and means of transport. The factors associated with hospitalization in public healthcare facilities included region, medical insurance, and the number of children.

Table 5 reports the multiple regression on the determinants of visiting primary healthcare settings. The results show that the effect of the inverse Mills ratio was positive, suggesting that the Heckman model was appropriate to adjust biased estimates. According to the outpatient outcome model, factors affecting receiving outpatient services in primary healthcare settings were transportation, medical costs, emergency treatment, residence, the number of children alive, caring for grandchildren, and health status. In the inpatient outcome model, hospitalization in primary healthcare settings was correlated with transportation, medical costs and self-payment, residence, income, the number of chronic diseases and ADL difficulty.

## 4. Discussion

### 4.1. Healthcare-Seeking Behavior among Chinese Elderly People from 2011 to 2015

The results from three nationally representative surveys indicate substantial improvements regarding access to healthcare services among Chinese elderly people since 2011. Both the use of outpatient and inpatient services went up by half during the five years. The data also identified certain features of elderly patients’ hospital choices. Over the 5 years, elderly people showed an overwhelming preference for public healthcare providers, which consistently composed over seven-tenths of the outpatient services and nine-tenths of the inpatient services. The largest decrease in public outpatient service use occurred between 2011 and 2013, which may be due to a set of policies reforming public hospitals and supporting non-public medical institutions. Undeniably, patients usually have diverse perceptions and expectations regarding the quality of healthcare services provided by facilities with inverse ownership. Public hospitals are believed to provide high-quality healthcare services, whereas other private contenders are often believed to provide low-quality services [13].

In addition, the percentage of older adults who visited a primary healthcare setting for outpatient care was more than half. However, when seeking inpatient care, almost 80% of patients turned to high-level hospitals. Moreover, the annual decreases in healthcare use can be attributed to decreases in primary healthcare use. This is in sharp contrast to the sustained use of public and large hospitals. Without policy enforcers or a general practitioner system, patients always tend to bypass primary care and visit high-level hospitals for specialized treatment and large reimbursements [35].

### 4.2. Predictors of Healthcare-Seeking Behavior

#### 4.2.1. Predisposing Characteristics

In Asian terms, the medical systems of China and South Korea are similar. Research on the medical behavior of patients in South Korea shows that urban and rural areas, income, education level, health status, medical insurance and so on all affect the choice of medical institutions. There are some similarities with the results of this study [36,37,38,39]. This study found significant differences in healthcare utilization and provider choices among elderly people with different characteristics. Females and the oldest people were found to use inpatient services more, which is possibly attributable in part to the worse health status and cautious health beliefs of females and in part to the increase in the number of health problems as their lifespan extends, which is called the ‘cost of success’ [21,40,41,42]. The urban elderly tended to choose public healthcare providers and higher-level hospitals compared with their rural counterparts, possibly due to a higher financial capability. The lower availability of private medical services in rural areas may be one of the reasons for this phenomenon. Moreover, there is evidence revealing that rural patients often choose primary care by default because they consider the costs and convenience more [43].

#### 4.2.2. Enabling Resources

Income has long been recognized as an important predictor of healthcare use, whether it is positive or negative. Wealthy people are unlikely to demonstrate healthcare underutilization; in fact, they spend more money on healthcare, yet people with a low income face greater barriers to accessing sufficient medical care [18,44,45]. Other empirical studies have suggested that a better health status of wealthy people might result in a lower likelihood of care use [46]. Based on the data of five Western European countries from the European health, aging and Retirement Survey (SHARE), they analyzed the relationship between income and healthcare for middle-aged and elderly people over 50 years old, and found that low-income groups are more likely to give up treatment [47]. In addition, Meyer et al. (2013) found that low income and no formal job will become obstacles to the utilization of medical services through a survey of six different countries or regions in the Asia Pacific region [48]. The evidence from Hong Kong, China shows that retired residents over 60 years old find it more difficult to get medical help due to lower income [48]. Our finding is in agreement with the former result, which indicates a positive correlation between income and healthcare use. In addition, having financial constraints was found to be a main barrier in Chinese elderly people healthcare needs turning into healthcare demands. During the course of the survey, we found that there was an abandonment of visits, and therefore asked the investigators why they gave up visits or gave up hospitalization, and the results of this study were further analyzed and found that among the respondents who reported unmet health needs, 16.05% did not seek outpatient care due to ‘having no money’, and 52.89% gave up hospitalization for the same reason. Even among those who had a chance to be hospitalized, nearly half left the hospital before recovering because of financial hardship.

The difference in healthcare utilization was found to be significant between individuals with and without medical insurance. Compared to the uninsured people, the older adults who participated in UEMI, government medical insurance, medical aid, private medical insurance and urban non-employed person’s medical insurance had a higher possibility of using outpatient and inpatient services. In China, these types of medical insurance have the highest level of reimbursement rates; therefore, the beneficiaries have a higher incentive to seek formal healthcare [20]. Other universal medical insurance schemes, such as NCMS and URMI, did not help older people access healthcare services, which was expected. This finding is consistent with previous literature [17]. Despite the gap in reimbursement rates among different types of medical insurance, L. Zhang, Wang, Qian, and Ni (2014) demonstrated that reimbursement increases cannot affect diabetics people’ choices among different healthcare institutions [49]. However, an earlier study indicated a strong connection between insurance type and the patient’s choice of medical provider, as Government and Labour Medical insurance beneficiaries preferred county hospitals, while those of the Cooperative Medical System used more village-level facilities than self-pay patients [23]. The study by C. Zhang et al. (2017) found that urban medical insurance membership increased the likelihood of using hospitals with higher quality but was not applicable for NCMS beneficiaries [20]. In our study, both medical insurance membership and different schemes had little influence on the elderly people’s choice of healthcare provider. As proposed by H. Wang, Zhang, Hou, Yan, and Hou (2018)’s study on internal migrants, our results indicate that universal medical insurance schemes improved the accessibility of healthcare to Chinese elderly people but rarely played a part in triaging patients and guiding healthcare behavior [50]. In addition, having a pension was also found to promote outpatient services utilization.

Influenced by Confucian ethics, intergenerational relationships among Chinese families are characterized by two-way transfers and filial piety. It is common that elderly people deplete their resources by giving them to their children and rely on them in their old age, while the adult children are expected to fulfil the filial responsibility to provide financial support and care for their parents in their old age [51]. In some cases, grandparents help perform child care duties and even continue to help their adult children financially [52]. There is a considerable amount of literature on the effect of intergenerational relationships on the health of elderly people compared to that on healthcare behavior. Similar to the idea proposed by B. Zhu and Mao (2017), our results indicate that financial support from children served as enabling resources and increased the possibility of seeking formal healthcare [53]. In contrast, giving adult children financial support exerted a crowding-out effect on healthcare utilization, which concurs well with a previous study [54].

Transportation barriers and high medical expenses have been shown to correlate with healthcare underutilization [55,56]. It is plausible that longer distances and higher charges deter patients or influence their choice of hospitals [57]. In this study, data on transportation availability and medical expenses were only observable when the elderly people reported healthcare utilization. In addition, there might exist causal interaction problems between the two factors and healthcare behavior. Therefore, the results should be interpreted with caution. We found that visits to public hospitals were more likely to occur through public transport. The reason is that public hospitals in China are funded and owned by the government and have more access to public resources. Therefore, they were more likely to be connected with public transport than were private hospitals. The same superiority can be found in terms of reimbursement coverage. Although the reimbursement rates are the same regardless of the ownership of hospitals, the proportion of insurance coverage for private providers is on average lower [13]. Our finding that the healthcare provided by public hospitals featured higher total medical expenses but lower Out-of-pocket portion (OOP) shares confirms this. As expected, primary healthcare facilities were found to have better transportation availability, and the healthcare received at the primary level was less expensive.

#### 4.2.3. Healthcare Need Factors

Prior research has generally concluded that healthcare use is primarily determined by the health status of individuals [58]. Our finding is consistent with other studies, as all need factors were strong determinants of healthcare-seeking behavior and had greater predictive value compared to the predisposing and enabling variables in Andersen’s model [16,29]. The likelihood of seeking formal healthcare was higher for the ‘not good’ health group than for the ‘good’ health group. The number of chronic diseases and ADL difficulty were also positively associated with healthcare utilization. A recent study in India found that chronically ill elderly people tended to avail public healthcare providers rather than private [59]. Our results do not appear to corroborate their observation, and they in fact confirm the results of Q. Wang et al. (2016)’s study, which suggested that health status did not significantly impact the choice of hospital ownership [13]. However, we found that elderly people with worse self-reported health and more chronic diseases were more likely to use primary outpatient care. In addition, chronic conditions and ADL difficulty increased the possibility of hospitalization in primary healthcare facilities. The reason is that poor health status leads to more healthcare needs and more frequent hospital visits; however, ADL difficulty restricts elderly people’s ability to navigate large hospitals. Furthermore, primary healthcare settings successfully provide affordable chronic disease management because financial constraints are a long-term consideration for these elderly people.

Last, when in need of emergency services, older adults had a strong preference for public and high-level healthcare providers. First, this result can be explained by the fact that a first-aid ambulance call to ‘120’ in China generally delivers patients to public hospitals. Second, together with the findings on medical expenses, it lends support to the descriptive results and reveals that older patients perceive public and high-level providers as more competitive and reliable, resulting in the under-utilization of private and primary healthcare services.

The current study is subject to several limitations. First, using cross-sectional data, we could not interpret the associations as causal associations. Longitudinal research is required to identify casualty. Second, CHARLS uses self-reported measures on health service utilization and the choice of healthcare providers, which are more prone to potential errors than objective measures. Future studies could use hospital registration information and clinical assessments to obtain more reliable results. Finally, we did not explore whether the healthcare-seeking behavior among Chinese older adults is reasonable, and the rationale of healthcare-seeking behavior among the Chinese elderly group should be explored in subsequent studies.

Even with these limitations, there are still some new findings in this study. Previous studies mainly focused on the impact of intergenerational economic support and intergenerational care on the physical and mental health of the elderly, but little attention was paid to its impact on healthcare-seeking behavior. Our results indicate that financial support from children served as enabling resources and increased the possibility of seeking formal healthcare. At the same time, the factors of medical needs have a significant impact on the choice of different levels of medical institutions, and the elderly people with worse self-reported health and more chronic diseases were more likely to use primary outpatient care. In addition, the study also found that the urban elderly tended to choose public healthcare providers and higher-level hospitals compared with their rural counterparts.

## 5. Conclusions

Our study detected changes in the patterns of healthcare-seeking behavior among Chinese elderly people from 2011 to 2015 and further explored the predictive factors. The proportion of elderly patients receiving outpatient and inpatient services increased steadily from 2011 to 2015, but the proportion of elderly patients who chose to receive outpatient and inpatient treatment in primary healthcare institutions decreased steadily. Public providers were consistently the first choice for individuals seeking formal healthcare, although the proportion of older adults who chose outpatient/inpatient admission to a public health facility all decreased in 2015 compared with 2011. The individuals who were female, younger and lived in urban, central or western regions, had medical insurance, had poor self-rated health and exhibited ADL impairment were more likely to use outpatient and inpatient services. The individuals who were female, oldest, rural residents, had poor self-rated health and exhibited ADL impairment relied more on private or primary medical institutions. In addition, transportation, medical expenses and the out-of-pocket ratio, as well as the urgency of the disease, were associated with the choice of provider. The universal medical insurance schemes improved health service utilization for the elderly population but had little impact on the choice of medical institution. In conclusion, elderly people’s healthcare-seeking behavior is the result of the integrated action of many internal causes and external conditions. It originates from health needs, is constrained by enabling resources, and is influenced by predisposing characteristics. Healthcare access has been improved; however, disorderly and unreasonable healthcare-seeking behavior is common among Chinese elderly people. Private and primary healthcare facilities are underutilized. More efforts should be devoted to encouraging reasonable healthcare-seeking behavior to improve the efficiency of diverse healthcare providers as a whole.

## Figures and Tables

**Figure 1 ijerph-18-02969-f001:**
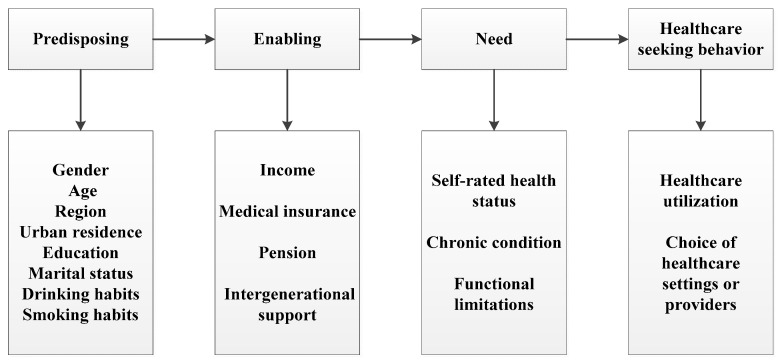
The Anderson theoretical model of influencing factors on healthcare-seeking behavior.

**Figure 2 ijerph-18-02969-f002:**
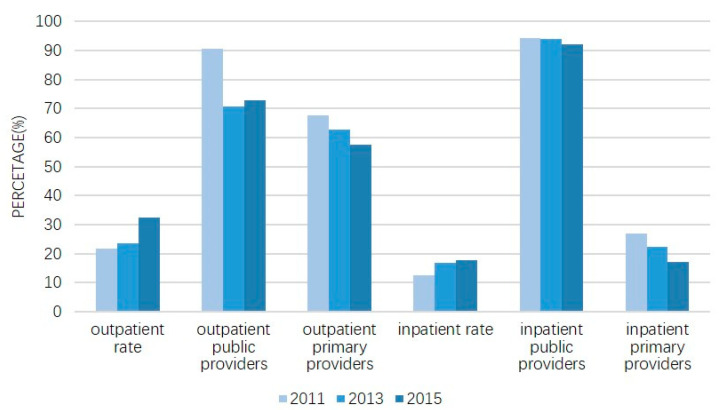
Outpatient and inpatient rates and the choice of healthcare providers from 2011–2015 (color should be used for any figures in print).

**Figure 3 ijerph-18-02969-f003:**
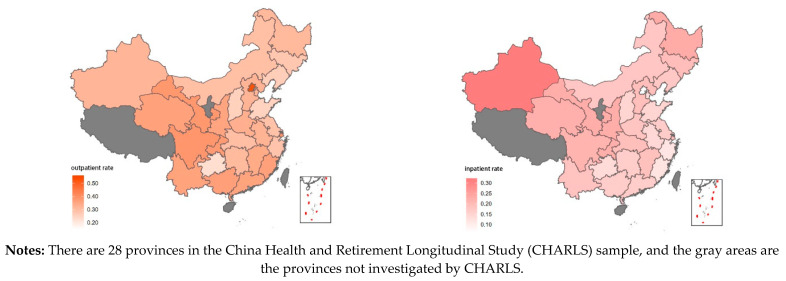
Geographic healthcare services utilization in China in 2015.

**Table 1 ijerph-18-02969-t001:** Sample characteristics by outpatient healthcare utilization and the choices of outpatient healthcare providers (*N* = 10,172).

Variable	Total	Outpatient (%)					
		Utilization	Non-Utilization	Public	Private	Primary	Higher-Level
Total		32.41	67.59	72.93	27.07	57.63	42.37
Gender							
Male	49.48	31.21 **	68.79 **	75.65 **	24.35 **	53.40 ***	46.6 ***
Female	50.52	33.55	66.45	70.62	29.38	61.23	38.77
Age							
60–79	89.57	32.24 **	67.76 **	72.88	27.12	57.01 ***	42.99 ***
≥80	10.43	35.80	64.2	72.57	27.43	65.34	34.66
Region							
East	31.32	30.56 ***	69.44 ***	80.12 ***	19.88 ***	54.26 ***	45.74 ***
Central	28.45	31.65	68.35	63.19	36.81	63.62	36.38
West	33.28	35.72	64.28	73.21	26.79	59.50	40.5
North-east	6.95	27.92	72.08	80.90	19.1	27.47	72.53
Residence							
Rural	71.08	31.85 *	68.15 *	69.44 ***	30.56 ***	66.82 ***	33.18 ***
Urban	28.92	33.82	66.18	82.31	17.69	32.75	67.25
Education							
No formal education	56.78	32.43	67.57	68.96 ***	31.04 ***	66.94 ***	33.06 ***
Elementary school	22.02	31.76	68.24	74.15	25.85	54.78	45.22
Middle school and above	21.20	33.90	66.1	82.08	17.92	39.76	60.24
Marital status							
Unmarried	21.42	35.04 ***	64.96 ***	68.09 ***	31.91 ***	62.63 **	37.37 **
Married	78.58	31.69	68.31	74.28	25.72	56.25	43.75
Medical insurance							
No insurance	10.58	27.45 ***	72.55 ***	72.41 ***	27.59 ***	58.33 ***	41.67 ***
UEMI	10.95	36.99	63.01	82.17	17.83	31.90	68.1
URMI	5.83	30.76	69.24	75.79	24.21	49.48	50.52
NCMS	66.08	31.96	68.04	69.37	30.63	65.44	34.56
Others	6.57	37.80	62.2	90.12	9.88	33.94	66.06
Self-reported health							
Not good	76.5	36.40 ***	63.60 ***	72.79	27.21	57.83	42.17
Good	23.5	18.23	81.77	76.05	23.95	60.74	39.26
ADL difficulty							
No	60.75	26.26 ***	73.74 ***	74.47 *	25.53 *	56.39 **	43.61 **
Yes	39.25	41.93	58.07	71.26	28.74	59.04	40.96

Note: UEMI—Urban Employee Medical Insurance; URMI—Urban Resident Medical Insurance; NCMS—New Rural Cooperative Medical Scheme; Chi square test is used for statistical analysis; *** *p* < 0.01, ** *p* < 0.05, * *p* < 0.1.

**Table 2 ijerph-18-02969-t002:** Sample characteristics by inpatient healthcare utilization and the choices of inpatient healthcare providers (*N* = 10,172).

Variable	Total	Inpatient (%)					
		Utilization	Non-Utilization	Public	Private	Primary	Higher-Level
Total		17.68	82.32	92.18	7.82	17.00	83
Gender							
Male	49.48	17.50	82.50	92.50	7.50	16.43	83.57
Female	50.52	17.83	82.17	91.86	8.14	17.57	82.43
Age							
60–79	89.57	22.63 ***	77.37 ***	91.73	8.27	16.44	83.56
≥80	10.43	17.23	82.77	94.71	5.29	21.39	78.61
Region							
East	31.32	15.22 ***	84.78 ***	94.33 ***	5.67 ***	12.72 ***	87.28 ***
Central	28.45	17.07	82.93	92.27	7.73	20.85	79.15
West	33.28	20.09	79.91	92.47	7.53	18.40	81.6
North-east	6.95	19.63	80.37	83.04	16.96	11.61	88.39
Residence							
Rural	71.08	16.85 ***	83.15 ***	93.01 *	6.99 *	21.65 ***	78.35 ***
Urban	28.92	19.78	80.22	90.46	9.54	7.14	92.86
Education							
No formal education	56.78	17.65	82.35	93.02	6.98	20.30 ***	79.70 ***
Elementary school	22.02	16.96	83.04	93.16	6.84	19.19	80.81
Middle school and above	21.20	18.57	81.43	89.24	10.76	9.00	91
Marital status							
Unmarried	21.42	20.20 ***	79.8 ***	92.96	7.04	19.83	80.17
Married	78.58	16.99	83.01	91.91	8.09	16.08	83.92
Medical insurance							
No insurance	10.58	15.97 ***	84.03 ***	96.12	3.88	14.06 ***	85.94 ***
UEMI	10.95	22.92	77.08	90.55	9.45	5.64	94.36
URMI	5.83	18.21	81.79	91.40	8.6	6.45	93.55
NCMS	66.08	16.78	83.22	92.51	7.49	22.20	77.8
Others	6.57	20.55	79.45	87.37	12.63	8.16	91.84
Self-reported health							
Not good	76.5	19.67 ***	80.33 ***	91.72	8.28	16.87	83.13
Good	23.5	9.32	90.68	92.31	7.69	16.94	83.06
ADL difficulty							
No	60.75	12.50 ***	87.5 ***	91.82	8.18	15.12 *	84.88 *
Yes	39.25	25.69	74.31	92.45	7.55	18.58	81.42

Note: Chi square test is used for statistical analysis. *** *p* < 0.01, * *p* < 0.1.

**Table 3 ijerph-18-02969-t003:** Results of the probit regression model on the determinants of healthcare utilization.

Variables	Outpatient		Inpatient	
	Coef.	S.E.	Coef.	S.E.
**Predisposing characteristics**				
Gender (ref: female)	−0.04	(0.04)	0.16 ***	(0.05)
Age (ref: 60–79)	−0.08	(0.06)	0.11 **	(0.06)
Region (ref: east)				
Central	−0.08 **	(0.04)	−0.01	(0.04)
West	0.02	(0.04)	0.10 **	(0.04)
North-east	−0.36 ***	(0.07)	0.08	(0.06)
Urban residence (ref: rural)	−0.05	(0.04)	0.04	(0.04)
Education (ref: no formal education)				
Elementary school	0.01	(0.04)	−0.01	(0.04)
Middle school and above	0.08 *	(0.04)	0.04	(0.05)
Married (ref: unmarried)	−0.02	(0.04)	−0.08 **	(0.04)
Smoking	−0.03	(0.04)	−0.02	(0.04)
Drinking	−0.03	(0.04)	−0.21 ***	(0.04)
Healthcare satisfaction	−0.01	(0.01)	−0.04 ***	(0.01)
**Enabling resources**				
Logpce	0.04 ***	(0.01)	0.05 ***	(0.01)
Medical insurance (ref: no insurance)				
UEMI	0.13 *	(0.07)	0.21 ***	(0.07)
URMI	−0.05	(0.08)	0.07	(0.08)
NCMS	0.08	(0.05)	0.06	(0.05)
Others	0.29 ***	(0.09)	0.08	(0.09)
Multiple medical insurances	−0.08	(0.09)	0.08	(0.09)
Pension	0.07 **	(0.03)	−0.02	(0.03)
Number of children alive	0.02 **	(0.01)	0.02 **	(0.01)
Number of children living together	0.02	(0.02)	−0.02	(0.02)
Economic support from children	0.01 **	(0.00)	0.01	(0.01)
Economic support to children	−0.01 **	(0.00)	0.00	(0.00)
Caring grandchildren	0.06 *	(0.03)	−0.06 *	(0.03)
**Need factors**				
Self-reported health (ref: not good)	−0.34 ***	(0.04)	−0.20 ***	(0.04)
Number of chronic diseases	0.13 ***	(0.01)	0.17 ***	(0.01)
ADL difficulty (ref: no)	0.14 ***	(0.03)	0.32 ***	(0.03)
Observations	10,164		10161	
Prob > chi2	<0.001		<0.001	
Log likelihood	−4928.20		−4305.50	
pseudo R-squared	0.12		0.09	

*** *p* < 0.01, ** *p* < 0.05, * *p* < 0.1.

**Table 4 ijerph-18-02969-t004:** Results of the probit regression model on the determinants of visiting public healthcare providers.

Variables	Outpatient		Inpatient	
	Coef.	S.E.	Coef.	S.E.
**Predisposing characteristics**				
Gender (ref: female)	0.10	(0.12)	−0.01	(0.24)
Age (ref: 60–79)	0.11	(0.16)	0.16	(0.32)
Region (ref: east)				
Central	−0.29 ***	(0.10)	−0.12	(0.23)
West	−0.04	(0.10)	−0.29	(0.23)
North-east	−0.07	(0.22)	−0.87 ***	(0.29)
Urban residence (ref: rural)	0.23 **	(0.11)	0.01	(0.21)
Education (ref: no formal education)				
Elementary school	−0.05	(0.10)	−0.01	(0.21)
Middle school and above	0.12	(0.12)	−0.22	(0.21)
Married (ref: unmarried)	0.20 **	(0.10)	0.29	(0.19)
Smoking	−0.02	(0.12)	−0.13	(0.22)
Drinking	−0.03	(0.10)	−0.01	(0.20)
Healthcare satisfaction	−0.06 *	(0.03)	−0.02	(0.07)
**Enabling resources**				
Logpce	−0.03	(0.03)	0.01	(0.06)
Medical insurance (ref: no insurance)				
UEMI	−0.03	(0.20)	−0.33	(0.44)
URMI	0.07	(0.21)	−0.46	(0.46)
NCMS	0.07	(0.15)	−0.29	(0.36)
Others	0.47 *	(0.27)	−1.19 **	(0.51)
Multiple medical insurances	0.11	(0.25)	0.59	(0.45)
Pension	0.12	(0.09)	−0.17	(0.19)
Number of children alive	−0.03	(0.03)	−0.15 ***	(0.05)
Number of children living together	−0.09 *	(0.05)	0.28 **	(0.12)
Economic support from children	0.00	(0.01)	0.00	(0.03)
Economic support to children	−0.00	(0.01)	−0.01	(0.02)
Caring grandchildren	−0.02	(0.08)	−0.03	(0.16)
One-way travel time	0.02	(0.02)	0.03	(0.07)
Means of transport (ref: others)				
Bus	0.21 *	(0.11)	0.32 *	(0.18)
Walk	−0.43 ***	(0.10)	0.22	(0.28)
Travel cost	−0.02	(0.04)	0.02	(0.06)
Total medical cost	0.10 ***	(0.03)	0.31 ***	(0.08)
Out-of-pocket portion	−1.04 ***	(0.15)	−0.22	(0.26)
**Need factors**				
Self-reported health (ref: not good)	0.12	(0.12)	0.05	(0.24)
Number of chronic diseases	0.00	(0.02)	0.03	(0.04)
ADL difficulty (ref: no)	−0.04	(0.08)	−0.04	(0.16)
Emergency	0.96 ***	(0.31)	—	—
Observations	1518		716	
Prob > chi2	<0.001		0.001	
Log likelihood	−756.41		−169.72	
pseudo R-squared	0.15		0.16	

*** *p* < 0.01, ** *p* < 0.05, * *p* < 0.1.

**Table 5 ijerph-18-02969-t005:** Results of the Heckman model on the determinants of visiting primary healthcare settings.

Variables	Outpatient Outcome Model		Outpatient Selection Model		Inpatient Outcome Model		Inpatient Selection Model	
	Coef.	S.E.	Coef.	S.E.	Coef.	S.E.	Coef.	S.E.
**Predisposing characteristics**								
Gender (ref: female)	0.01	(0.09)	0.01	(0.05)	0.07	(0.12)	0.19 ***	(0.05)
Age (ref: 60–79)	−0.02	(0.12)	−0.14 **	(0.07)	0.26 *	(0.15)	0.07	(0.07)
Region (ref: east)								
Central	−0.00	(0.07)	−0.04	(0.04)	0.22 *	(0.11)	0.06	(0.05)
West	0.04	(0.07)	0.07 *	(0.04)	0.17	(0.11)	0.15 ***	(0.05)
North-east	−0.38 ***	(0.14)	−0.33 ***	(0.08)	0.18	(0.18)	0.20 ***	(0.07)
Urban residence (ref: rural)	−0.6 2 ***	(0.11)	−0.09 **	(0.04)	−0.41 ***	(0.13)	0.10 **	(0.05)
Education (ref: no formal education)								
Elementary school	−0.05	(0.07)	0.04	(0.04)	0.12	(0.10)	−0.01	(0.05)
Middle school and above	−0.08	(0.08)	0.09 *	(0.05)	−0.11	(0.12)	0.08	(0.05)
Married (ref: unmarried)	0.05	(0.07)	0.01	(0.04)	0.01	(0.10)	−0.08 *	(0.05)
Smoking	−0.14 *	(0.08)	−0.06	(0.05)	0.00	(0.11)	−0.01	(0.05)
Drinking	0.06	(0.07)	−0.02	(0.04)	−0.05	(0.11)	−0.22 ***	(0.05)
Healthcare satisfaction	0.01	(0.02)	−0.00	(0.01)	0.01	(0.04)	−0.03 **	(0.02)
**Enabling resources**								
Logpce	0.01	(0.02)	0.06 ***	(0.01)	0.08 **	(0.03)	0.10 ***	(0.02)
Medical insurance (ref: no insurance)								
UEMI	0.04	(0.14)	0.17 **	(0.08)	0.07	(0.22)	0.23 ***	(0.09)
URMI	0.02	(0.15)	0.08	(0.09)	−0.06	(0.25)	0.13	(0.10)
NCMS	0.18 *	(0.10)	0.17 ***	(0.06)	0.23	(0.16)	0.13 **	(0.07)
Others	−0.13	(0.19)	0.31 ***	(0.10)	−0.28	(0.30)	−0.01	(0.12)
Multiple medical insurances	0.19	(0.17)	−0.05	(0.09)	0.37	(0.26)	0.15	(0.11)
Pension	0.00	(0.06)	0.07 **	(0.04)	−0.05	(0.09)	0.01	(0.04)
Number of children alive	0.07 ***	(0.02)	0.02	(0.01)	0.02	(0.03)	0.02	(0.01)
Number of children living together	0.06 *	(0.04)	0.01	(0.02)	0.01	(0.05)	−0.01	(0.02)
Economic support from children	0.01	(0.01)	0.01 **	(0.01)	0.02	(0.01)	0.01 **	(0.01)
Economic support to children	−0.01	(0.01)	−0.01	(0.00)	0.01	(0.01)	0.00	(0.00)
Caring grandchildren	0.11 **	(0.05)	0.06 *	(0.03)	−0.03	(0.09)	−0.02	(0.04)
One-way travel time	−0.02	(0.01)	—	—	−0.01	(0.02)	—	—
Means of transport (ref: others)								
Bus	−0.40 ***	(0.09)	—	—	−0.25 ***	(0.09)	—	—
Walk	0.27 ***	(0.08)	—	—	0.08	(0.12)	—	—
Travel cost	−0.13 ***	(0.03)	—	—	−0.13 ***	(0.03)	—	—
Total medical cost	−0.18 ***	(0.03)	—	—	−0.39 ***	(0.07)	—	—
Out-of-pocket portion	−0.11	(0.08)	—	—	−0.32 **	(0.12)	—	—
**Need factors**								
Self-reported health (ref: not good)	−0.34 ***	(0.08)	−0.35 ***	(0.04)	0.04	(0.12)	−0.18 ***	(0.05)
Number of chronic diseases	0.12 ***	(0.02)	0.13 ***	(0.01)	0.11 ***	(0.02)	0.15 ***	(0.01)
ADL difficulty (ref: no)	0.06	(0.06)	0.05	(0.04)	0.19 **	(0.08)	0.26 ***	(0.04)
Emergency	−0.52 ***	(0.16)	—	—	—	—	—	—
Observations	9543				9418			
Inverse Mills ratio	1.58 ***		(0.34)		1.32 ***		(0.34)	
Prob > chi2	<0.001				0.001			
Log likelihood	−4517.29				−3350.70			

*** *p* < 0.01, ** *p* < 0.05, * *p* < 0.1.

## Data Availability

The CHARLS dataset is publicly available. Information about the data source and available data are found at http://charls.pku.edu.cn/pages/data/111/zh-cn.html (accessed on 17 February 2021).

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
