# Peer review of "Healthcare-Seeking Behavior among Chinese Older Adults: Patterns and Predictive Factors"

_ijerph, 2021, doi:10.3390/ijerph18062969_

Round 1

Reviewer 1 Report

The objective of the present manuscript was to investigate the patterns and predictive factors of healthcare seeking behavior among older Chinese adults.

I believe the necessary modifications were made for a better understanding of the manuscript, only some minor observations remain

  • In line 244 it mentions that "almost half of the elderly people received outpatient care in primary facilities", however figure 1 shows that in all years this percentage was greater than 50%
  • Table 1 and 2 do not mention the statistical test used, there are too many asterisks representing the same thing: the comparison between two groups, for example an * in the “private” group is no longer necessary if an * was placed in the “public” group, unless these asteristics represent another comparison that is not mentioned.
  • In the discussion on lines 378-379, does “unmet health needs” refers to “Healthcare satisfaction” or “Self-reported health as not good”?, in any case other results are presented that had not been previously mentioned as' having no money ".
  • The discussion is mainly focused on comparing with studies from China, only rarely are some variables compared with other populations but also the possible causes of the differences are not explained, for example, which characteristics does the Chinese population have versus the population of India that explains chronically ill elderly people tended to avail public healthcare providers rather than private ?, Comparisons with other populations would be of primary interest to readers outside of China.
  • Review the format of the bibliography in the main text 

Author Response

  1. In line 244 it mentions that "almost half of the elderly people received outpatient care in primary facilities", however figure 1 shows that in all years this percentage was greater than 50%  

Response:

Thank you for your careful thoughts. Per your suggestion, we have changed "almost half of the elderly people received outpatient care in primary facilities" to "more than half of the elderly people received outpatient care in primary facilities" (p6 line251)

  1. Table 1 and 2 do not mention the statistical test used, there are too many asterisks representing the same thing: the comparison between two groups, for example an * in the “private” group is no longer necessary if an * was placed in the “public” group, unless these asteristics represent another comparison that is not mentioned.

Response:

Thank you for your kind minds. We have commented on the statistical method in the table, as showed in p8 line 271 and line 276:

“Note: Chi square test is used for statistical analysis.”

In Tables 1 and 2, asterisk does not represent comparison between the two groups, but rather represents the comparison within the group, such as the difference between male and female in the "private" group and the difference between male and female in the "public" group.

  1. In the discussion on lines 378-379, does “unmet health needs” refers to “Healthcare satisfaction” or “Self-reported health as not good”? in any case other results are presented that had not been previously mentioned as' having no money ".

 Response:

Thank you for your careful thoughts. Unmet health needs refer to the reasons for not seeing a doctor or going to hospital. During the course of the survey, we found that there was an abandonment of visits, and therefore asked the investigators why they gave up visits or gave up hospitalization.

In order to make “Unmet health needs” easier to understand, we have added the following contents:

“During the course of the survey, we found that there was an abandonment of visits, and therefore asked the investigators why they gave up visits or gave up hospitalization, and the results of this study were further analyzed and found that among the respondents who reported unmet health needs, 16.05% did not seek outpatient care due to ‘having no money’, and 52.89% gave up hospitalization for the same reason. Even among those who had a chance to be hospitalized, nearly half left the hospital before recovering because of financial hardship.” (p15 line397-400)

  1. The discussion is mainly focused on comparing with studies from China, only rarely are some variables compared with other populations but also the possible causes of the differences are not explained, for example, which characteristics does the Chinese population have versus the population of India that explains chronically ill elderly people tended to avail public healthcare providers rather than private ?, Comparisons with other populations would be of primary interest to readers outside of China.

Response:

Thank you for your careful thoughts. Per your suggestion, we have added some contents which are compared with those of other countries in the discussion part of this paper, as follows:

“In Asian countries, the medical systems of China and South Korea are similar. Research on the medical behavior of patients in South Korea shows that urban and rural areas, income, education level, health status, medical insurance and so on all affect the choice of medical institutions. There are some similarities with the results of this study (Kim A M et al.,2018; Kim B R,1990; Lee J C et al.,2011; You C H& Kwon Y D ,2012).” (p14 line 366-370)

“Based on the data of five Western European countries from the European health, aging and Retirement Survey (SHARE), they analyzed the relationship between income and health care for middle-aged and elderly people over 50 years old, and found that low-income groups are more likely to give up treatment (Mielk et al.,2009). In addition, Meyer et al. (2013) found that low income and no formal job will become obstacles to the utilization of medical services through a survey of six different countries or regions in the Asia Pacific region. The evidence from Hong Kong, China shows that retired residents over 60 years old are more difficult to get medical help due to lower income (Meyer et al., 2013).” (p15 line387-394)

  1. Review the format of the bibliography in the main text 

Response:

Thank you for your kind minds. We have revised the references in the main text.

Reviewer 2 Report

This study reports findings of a descriptive statistical analysis of the patterns and predictive factors of healthcare seeking behavior among older Chinese adults using sample survey data from 10914 participants aged ≥60 years from the 2011, 2013 and 2015 China Health and Retirement Longitudinal Study. Overall, the analyses appear sound.

The English grammar of the text is reasonably good, but can benefit from a moderate edit.

The text also needs to be carefully studied to make sure that terminology is clearly defined and articulated and similarly the results of the statistical analyses.

Some specific examples of items that need clarity and more detail:

-- Andersen's behavioral model of health service use is first referenced in the paragraph just before Section 2; it would be good to state more explicitly the components of this model and how they are related. For example, this could be done in a graph, and you could highlight the segments of the overall model on which you focus in the paper.

-- In the titles of Tables 3, 4, and 5, you need to make clear that these are the estimates of a specific category of regression models, namely probit regression models, and for each regression model you need to make clear how the outcome/dependent variables are scored/measured. 

-- The discussion of the statistical findings in Section 4 generally are good. One question occurs to me: Generally, your findings of the statistically significant regressors are consistent with those of other studies and thereby expected and reasonable. But are there any particular findings that stand out as perhaps not to be expected from prior studies and that thereby merit additional comment and perhaps further research?

Author Response

请参阅附件。

Round 2

Reviewer 2 Report

The revisions in response to the previous review have improved the manuscript substantially. Very good.

However, the manuscript needs one more final check of the grammar to correct/polish a few places. For example:

-- on line 153 the word "basic" in "On the basic of the Anderson behavior model" should be "basis".

-- on line 402 the word "a" in "we found that there was a abandonment of visits" should be "an".

This manuscript is a resubmission of an earlier submission. The following is a list of the peer review reports and author responses from that submission.

Round 1

Reviewer 1 Report

This was a very difficult study to understand for a reviewer not familiar with the Chinese healthcare landscape.  The authors need to decide the target audience for this work.  If it is an international audience, more explanation is required of the current situation in China and why this may be relevant to other countries.  If the target audience is domestic, I would recommend a Chinese language journal.

Perhaps due to my lack of understanding of healthcare in China, the importance of the comparisons and interrelationships along three axes public vs. private utilization, primary vs. "higher-level," and inpatient vs. outpatient was not clear to me.  It might help if clearer hypotheses or questions with regard to these distinctions were set forth early in the manuscript.

The analytic model is not adequately described in section 2.3 for me (a statistician) to clearly understand the methodology.  How does equation (or stage) 1 relate mathematically to equation (or stage) 2?  

A number of large tables are presented with many factors.  The utility of these results in forming an understanding of major forces driving health care utilization in China and any change between 2011 and 2015 are not clear to me.  Perhaps some major themes could be hypothesized and then rejected or accepted based on the results.  I could not piece together the results in any cohesive narrative.

There seem to be two major conclusions.  First, a trend to increasing utilization.  Second, "disorderly and unreasonable healthcare seeking behaviour is common among Chinese elderly 
people."  This second conclusion is difficult to understand since all the factors demonstrated to explain utilization seem rather logical from the consumers' point of view.  I am wondering if the authors have in mind a construct or model for what an efficient healthcare system in China would look like in 2015. Perhaps then the data demonstrates a pattern not matching that ideal model.  If this is the case, stating the ideal model at the beginning of the manuscript and then using the data to demonstrate deviation from this ideal might provide an overarching narrative and clearer set of actionable conclusions from the analysis of the data.  

Author Response

Manuscript ID: ijerph-1007083

Title: Healthcare Seeking Behavior among Chinese Older Adults: Patterns and Predictive Factors

We gratefully thank the editor and all reviewers for their time spend making their constructive remarks and useful suggestion, which has significantly raised the quality of the manuscript and has enable us to improve the manuscript. Each suggested revision and comment, brought forward by the reviewer was accurately incorporated and considered. Below the comments of the reviewer are response point by point and the revisions are indicate:

  1. This was a very difficult study to understand for a reviewer not familiar with the Chinese healthcare landscape.  The authors need to decide the target audience for this work.  If it is an international audience, more explanation is required of the current situation in China and why this may be relevant to other countries.  If the target audience is domestic, I would recommend a Chinese language journal.

Response:

Thank you for your careful thoughts. With this paper, we try to contribute to a better understanding of the health care utilization in developing countries. At the same time, we also add descriptions in the background part of the manuscript focusing on China's medical reform and current status as following:

“In the early years, China's health care system has been provided by hospitals, especially public hospitals, with 90% of the total number of in-patient and out-patient medical services (Zhu, Y, 2013)” (p2 line65-66)

“After several rounds of reform were implemented, China achieved large improvements in access to healthcare services, the private health care sector has continued to expand in China. However, unexpectedly, although private facilities have been estimated to constitute 46.7% of all health care facilities nationwide, the share of health services provided by private providers was determined to have only increased to approximately 10% in 2013 (Health and Family Planning Commission, 2015).” (p2 line 68-72)

“In principle, patients in China can walk in and obtain treatment at any level of health care facility of any ownership without restrictions, especially in the case of outpatient care services (Karen Eggleston, 2010) . The reason why private health services are not growing as fast as private health care facilities remains unknown. This has led to increasing interest in patient choice between private and public providers in China (Wang Chen, 2013).” (p2-3 line93-98)

  1. Perhaps due to my lack of understanding of healthcare in China, the importance of the comparisons and interrelationships along three axes public vs. private utilization, primary vs. "higher-level," and inpatient vs. outpatient was not clear to me.  It might help if clearer hypotheses or questions with regard to these distinctions were set forth early in the manuscript.

Response:

Thank you for your kind minds. Per your suggestion, we added the definitions of relevant outcome variables such as public vs. private utilization, primary vs. higher level, and inpatient vs. outpatient in materials and methods, as showed in p4 line 150-160:

“Reporting healthcare utilization is measured by (1) visiting a doctor for outpatient care in the last month; (2) receiving inpatient care in the past year. Choice of healthcare providers is measured by (1) whether the healthcare provider the respondent visited most recently for outpatient care during the last month / or for hospitalization in the past year was public or private, among which, public hospitals refer to the hospitals organized by the government and included in financial budget management; private hospitals refer to non-governmental public hospitals with private nature. In China, most of the private hospitals are health institutions funded by the society and run by profit-making organizations; there are also a few non-profit organizations that enjoy government subsidies.; (2) whether the healthcare provider above was a primary healthcare facility, which comprises community healthcare centers, township hospitals, healthcare posts and village clinics. All outcome variables are coded as 1 if yes, 0 otherwise.”

  1. The analytic model is not adequately described in section 2.3 for me (a statistician) to clearly understand the methodology.  How does equation (or stage) 1 relate mathematically to equation (or stage) 2?  

Response:

Thank you for your kind minds. Per your suggestion, we added a detailed description of the statistical model in the analysis section,as showed in p5 line 205-209:

“Where is a latent variable of underlying probability of reporting healthcare utilization, is the elderly’s choice of healthcare providers. And  is observable only when an old adult used healthcare services (=1) and  are the vectors of factors predicting healthcare utilization decision and the choice of healthcare provider. is the inverse Mills ratio. and  are constant terms,and are error terms.”

  1. A number of large tables are presented with many factors.  The utility of these results in forming an understanding of major forces driving health care utilization in China and any change between 2011 and 2015 are not clear to me.  Perhaps some major themes could be hypothesized and then rejected or accepted based on the results.  I could not piece together the results in any cohesive narrative.

Response:

Thank you for your careful thoughts. Based on the Andersen’s behavioral model of health service use, this study analyzes the patterns and factors of healthcare seeking behavior among Chinese elderly people from three dimensions: predisposing characteristic, enabling resources, and health care needs. The predisposing characteristics were sex, age, region, urban residence, educational background, marital status, drinking habits, and smoking habits, and the enabling resources consisted of income, medical insurance, pension and intergenerational support. The need factors were self-rated health status, chronic condition, and functional limitations. Tables 2 to 4 show the influence of the above factors on whether older adults choose medical treatment, the choice of healthcare providers, and whether they choose primary medical institutions, respectively.

The changes of healthcare seeking behavior among Chinese elderly from 2011 to 2015 are shown in Figure 1. It can be seen from the figure that the proportion of elderly patients receiving outpatient and inpatient services increased steadily from 2011 to 2015, but the proportion of elderly patients who chose to receive outpatient and inpatient treatment in primary health care institutions decreased steadily. Public providers were consistently the first choice for individuals seeking formal healthcare, although the proportion of older adults who chose outpatient / inpatient admission to a public health facility all decreased in 2015 compared with 2011. A detailed description of Figure 1 is given in lines 223 to 238.

  1. There seem to be two major conclusions.  First, a trend to increasing utilization.  Second, "disorderly and unreasonable healthcare seeking behavior is common among Chinese elderly 
    people."  This second conclusion is difficult to understand since all the factors demonstrated to explain utilization seem rather logical from the consumers' point of view.  I am wondering if the authors have in mind a construct or model for what an efficient healthcare system in China would look like in 2015. Perhaps then the data demonstrates a pattern not matching that ideal model.  If this is the case, stating the ideal model at the beginning of the manuscript and then using the data to demonstrate deviation from this ideal might provide an overarching narrative and clearer set of actionable conclusions from the analysis of the data.  

Response:

Thank you for your kind reminds. The purpose of this study is to detect changes in the patterns of healthcare seeking behavior among Chinese elderly people from 2011 to 2015 and further explored the predictive factors. We find that the individuals who were female, were younger, lived in urban, central or western regions, had medical insurance, had poor self-rated health and exhibited ADL impairment were more likely to use outpatient and inpatient services. The individuals who were female, were oldest-old, were rural residents, had poor self-rated health and exhibited ADL impairment relied more on private or primary medical institutions. The healthcare seeking behavior characteristics of the Chinese elderly population found in this study fit the Andersen’s behavioral model of health service use. However, this study did not explore whether this healthcare seeking behavior among Chinese older adults is reasonable, we have added notes in the limitations section of the manuscript. We will analyze the rationality of medical treatment behaviors among Chinese older adults in future studies, as showed in p14 line 443-445:

“Finally, we did not explore whether the healthcare seeking behavior among Chinese older adults is reasonable, the rational of healthcare seeking behavior among the Chinese elderly group should be explored in subsequent studies.”

Reviewer 2 Report

The present manuscript aims to
investigate the patterns and factors explaining healthcare seeking behavior among Chinese elderly people focused on two categories of behaviors: healthcare utilization and provider choices.
A descriptive analysis of three waves and an analysis of prognostic factors from a single wave are presented.
While an interesting topic, the data presented as is does not make a convincing and there are several concerns which need to be addressed.
Major Comments

It is not clear if the sample size (line 132) is the same for the 3 years analyzed

The acronym ADLs or IADLs must be defined and the measuring instrument must be indicated

It is not clear what is meant by primary providers, can a primary provider be both public and private? The characteristics of the providers must be defined in the methods

It is not described what type of ensurace each one is: granted by the government, participant self-paid

It is not clear how a patient can decide between seeking health-care in higher-levels hospitals or primary provider since in general it is assumed that this depends on how the health system is working and on the severity of the disease. Perhaps this decision may only be carried out  in those who seek health-care from private providers, while those who seek care from public providers may not have the opportunity to decide? Wouldn't the predictors involved be the same then?, also it is difficult to understand that a person can be hospitalized in a primary care service.

Several tables and figures are not properly referenced in the text by mistake

Table 1 refers only to the year 2015? It is not specified which are the comparisons made: public vs primary? between inpatient vs outpatient?

On line 213 it should say EUMI

Line 214 the acronyms URMI, NCMS must be defined

Figure 2 the gray color must be defined

In lines 259-266 the results of the outcome model are mentioned, I consider that the results obtained with the model selection should be highlighted since they are the adjust biased estimates

The discussion (L299-L302) does not consider the possibility that in rural areas the availability of private health services is lower than in urban areas and the differences are due to this and not to cost.

Patients with insurance would be expected to seek more health-care from private providers but no difference was found, it would be important to discuss the reason

L362 OOP must be defined

As in the results, the lack of clarity in the definitions of health services makes it difficult to understand the discussion, the comparison between seeking or not seeking health-care is clear, more or less clear between public and private as well as inpatient and outpatient but it is not clear between primary to higher level; how can an inpatient care be requested in a primary care service?

The conclusion in very general, and does not respond punctually to the objectives set out in the study, I consider the main changes in seeking health-care behaviour in the three waves and the factors associated with seeking care in health services should be pointed out.

The presentation of the writing seems unattending: the references are presented without uniformity, there are misspelled words, several undefined acronyms and the figures and tables appear as errors in the text.

It is presented as a limitation that a cross-sectional study was carried out instead of a longitudinal one, so that the associations found cannot be interpreted as causal associations. However, there were data from two other waves of the survey, why was a longitudinal design not used then?

Author Response

Manuscript ID: ijerph-1007083

Title: Healthcare Seeking Behavior among Chinese Older Adults: Patterns and Predictive Factors

We gratefully thank the editor and all reviewers for their time spend making their constructive remarks and useful suggestion, which has significantly raised the quality of the manuscript and has enable us to improve the manuscript. Each suggested revision and comment, brought forward by the reviewers was accurately incorporated and considered. Below the comments of the reviewers are response point by point and the revisions are indicate:

1.It is not clear if the sample size (line 132) is the same for the 3 years analyzed

Response:

Thank you for your kind reminds. We use the data of 2011, 2013 and 2015 to analyze the change trend of medical seeking behavior of the elderly in China, and further analyze the influencing factors of medical seeking behavior of the elderly in 2015. A total of 10172 respondents in the 2015 wave of CHARLS were finally included in this study. (p3-4 line 140-147)

2.The acronym ADLs or IADLs must be defined and the measuring instrument must be indicated

Response:

Thank you for your kind reminds. Per your suggestion, we defined ADLs or IADLs in the manuscript and indicated the methods of measurement, as showed in p4 line 176-181:

“Functional limitations are assessed with activities of daily living (ADLs) and instrumental activities of daily living (IADLs). ADLs involve six basic tasks: dressing, bathing or showering, eating, getting into or out of bed, toileting, and controlling urination and defecation. IADLs are composed of five higher ordered daily tasks: doing household chores, preparing hot meals, grocery shopping, managing money, taking medications. Having difficulty with any ADL or IADL because of health and memory problems is coded as 1, 0 otherwise.”

3.It is not clear what is meant by primary providers, can a primary provider be both public and private? The characteristics of the providers must be defined in the methods

Response:

Thank you for your kind reminds. Per your suggestion, we added the definitions of relevant outcome variables such as health utilization (inpatient or outpatient), public vs. private utilization, primary vs. higher level in materials and methods, as showed in p4 line 150-160:

Reporting healthcare utilization is measured by (1) visiting a doctor for outpatient care in the last month; (2) receiving inpatient care in the past year. Choice of healthcare providers is measured by (1) whether the healthcare provider the respondent visited most recently for outpatient care during the last month / or for hospitalization in the past year was public or private, among which, public hospitals refer to the hospitals organized by the government and included in financial budget management; private hospitals refer to non-governmental public hospitals with private nature. In China, most of the private hospitals are health institutions funded by the society and run by profit-making organizations; there are also a few non-profit organizations that enjoy government subsidies.; (2) whether the healthcare provider above was a primary healthcare facility, which comprises community healthcare centers, township hospitals, healthcare posts and village clinics. All outcome variables are coded as 1 if yes, 0 otherwise.”

4.It is not described what type of insurance each one is: granted by the government, participant self-paid

Response:

Thank you for your kind reminds. During the most recent health care reform in 2009, China's central government established multiple initiatives to further relax constraints on the growth of the private health care sector. These initiatives included the following: private health care providers were eligible to contract with public health insurance programs on the same terms as public providers(Central Committee of the Communist Party and State Council, 2009; Xu et al., 2015; Yip and Hsiao,2014).Both public and private hospitals can use medical insurance to seek medical care, but there is a difference in the reimbursement proportion, with public hospitals having national financial subsidies, and the reimbursement proportion will be higher than that of private hospitals. We describe each type of insurance in the text, as showed in p4 line 172-175:

“Medical insurance is measured by having any type of medical insurance (0=no insurance, 1=UEMI, Urban Employee Medical Insurance, 2=URMI, Urban Resident Medical Insurance, 3=NCMS, New Rural Cooperative Medical Scheme, 4=other types of insurance).”

  1. It is not clear how a patient can decide between seeking health-care in higher-levels hospitals or primary provider since in general it is assumed that this depends on how the health system is working and on the severity of the disease. Perhaps this decision may only be carried out in those who seek health-care from private providers, while those who seek care from public providers may not have the opportunity to decide? Wouldn't the predictors involved be the same then? also it is difficult to understand that a person can be hospitalized in a primary care service.

Response:

Thank you for your careful thoughts. Unlike some countries that have a gatekeeper system with mandatory referral, there has been limited research on options between different tiers of health care facilities. The medical care environment in China is relatively free. Patients are free to choose medical hospitals, and primary care service can also provide inpatient services. We add a few additions to this in Introduction:

“In principle, patients in China can walk in and obtain treatment at any level of health care facility of any ownership without restrictions, especially in the case of outpatient care services (Eggleston et al., 2010). The reason why private health services are not growing as fast as private health care facilities remains unknown. This has led to increasing interest in patient choice between private and public providers in China (Wang et al., 2013).” (p2-3 line93-98)

  1. Several tables and figures are not properly referenced in the text by mistake

Response:

Thank you for pointing out our mistake. We have made changes to the mistakes in the manuscript.

  1. Table 1 refers only to the year 2015? It is not specified which are the comparisons made: public vs primary? between inpatient vs outpatient?

Response:

Thank you for your careful thoughts. We divide the current situation of healthcare seeking behavior of the elderly in China into two parts: outpatient and inpatient, and we analyzed the current situation from three aspects: utilization rate, whether the medical institutions are public hospitals and whether the medical institutions are primary hospitals. For example, in terms of outpatient services, we studied whether the Chinese elderly had outpatient medical services in the past month, and the outpatient institutions were public or private, were primary or higher level; in terms of hospitalization, we have studied whether the elderly in China have used inpatient medical services in the past year,and the inpatient institutions were public or private, were primary or higher level. In order to simplify the table, Table 1 only shows the utilization rate of outpatient and inpatient medical services and the proportion of elderly people choosing public and primary medical institutions in 2015, and what's left is the proportion of the corresponding group.

  1. On line 213 it should say EUMI

Response:

Thank you for pointing out our mistake. We have the former wrong expressions carefully revised. The "URMI" has been changed to "UEMI". (p7 line257)

  1. Line 214 the acronyms URMI, NCMS must be defined

Response:

Thank you for your kind reminds. We have defined URMI and NCMS in the manuscript, as showed in p4 line 172-175:

“Medical insurance is measured by having any type of medical insurance (0=no insurance, 1=UEMI, Urban Employee Medical Insurance, 2=URMI, Urban Resident Medical Insurance, 3=NCMS, New Rural Cooperative Medical Scheme, 4=other types of insurance).”

  1. Figure 2 the gray color must be defined

Response:

Thank you for your kind reminds. We have defined the grey part in the manuscript. (p7 line266-267)

“There are 28 provinces in the CHARLS sample, and the gray areas are the provinces not investigated by CHARLS”

  1. In lines 259-266 the results of the outcome model are mentioned, I consider that the results obtained with the model selection should be highlighted since they are the adjust biased estimates

Response:

Thank you for your kind reminds. There is a distinction between observable and unobserved values, which can produce bias if statistical analysis is performed by observable values only. When an old adult is in need of healthcare, they make a decision to utilize healthcare services or not. Diverse obstacles might keep them from seeking healthcare, making their preference or choice of healthcare providers unobservable. If we simply ignore the sample selection bias and estimate Probit models using the nonrandom subsample (those who utilized healthcare services), biased estimates might occur. Thus we adopts the approach of the Heckman selection model to test and account for possible selection issue. We added description to the Heckman model in the analysis as following:

“The sample selection often arises from practice because of the partial observability of the outcome variable. When sample selection is present, the observed data do not represent a random sample of the population, even after checking explanatory variables, that is, the data missing in the sample does not respond to a random selection process(Zuzana Sarvašová, 2018). Thus, a standard analysis using only complete cases will lead to biased results. In order to overcome the possible sample selection bias and obtain the unbiased estimation of medical treatment behavior, we use the sample selection model proposed by James Joseph Heckman.” (p4 line 185-191)

  1. The discussion (L299-L302) does not consider the possibility that in rural areas the availability of private health services is lower than in urban areas and the differences are due to this and not to cost.

Response:

Thank you for your careful thoughts. Per your suggestion, supplementary discussion of selection differences by type of medical facility for urban and rural elderly has been presented in the latest manuscript. As follows:

Urban elderly tended to choose public healthcare providers and higher-level hospitals compared with their rural counterparts, possibly due to a higher financial capability. The lower availability of private medical services in rural areas may be one of the reasons for this phenomenon. Moreover, there is evidence revealing that rural patients often choose primary care by default because they consider the costs and convenience more (Yun Liu, Zhong, Yuan, & van de Klundert, 2018).”(p12 line 346-350)

  1. Patients with insurance would be expected to seek more health-care from private providers but no difference was found, it would be important to discuss the reason

Response:

Thank you for your careful thoughts.

This study mainly focuses on the impact of medical insurance on the elderly's healthcare services utilization, but whether there are medical insurance and the type of medical insurance have very little impact on the choice of medical institutions for the elderly. Compared with the elderly without medical insurance, enrollees with other medical insurance such as commercial medical insurance have a low probability of hospitalization to public medical institutions, and they are more likely to go to private medical institutions. This is also supported by the conclusions of other studies in China, such as Wang et al.'s (2018) study on an ambulatory population, which proved that the basic health insurance system improved the utilization probability of inpatient services but hardly played a role in guiding patients' rational medical treatment behaviors, only about 10% of the 6121 individuals covered in the study were hospitalized at primary level, and the remaining nine were half concentrated at secondary and tertiary hospital.

  1. L362 OOP must be defined

Response:

Thank you for your kind reminds. We have defined the OOP in the manuscript. (p13 line 410)

  1. As in the results, the lack of clarity in the definitions of health services makes it difficult to understand the discussion, the comparison between seeking or not seeking health-care is clear, more or less clear between public and private as well as inpatient and outpatient but it is not clear between primary to higher level; how can an inpatient care be requested in a primary care service?

Response:

Thank you for your careful thoughts. In China, primary care service can also provide inpatient services. Besides, we added the definitions of primary and higher level, as showed in p4 line158-160:

(2) whether the healthcare provider above was a primary healthcare facility, which comprises community healthcare centers, township hospitals, healthcare posts and village clinics. All outcome variables are coded as 1 if yes, 0 otherwise.”

  1. The conclusion in very general, and does not respond punctually to the objectives set out in the study, I consider the main changes in seeking health-care behavior in the three waves and the factors associated with seeking care in health services should be pointed out.

Response:

Thank you for your suggestive comment. Per your suggestion, we give a supplementary description of the conclusion, as showed in p14 line 448-461:

“The proportion of elderly patients receiving outpatient and inpatient services increased steadily from 2011 to 2015, but the proportion of elderly patients who chose to receive outpatient and inpatient treatment in primary health care institutions decreased steadily. Public providers were consistently the first choice for individuals seeking formal healthcare, although the proportion of older adults who chose outpatient / inpatient admission to a public health facility all decreased in 2015 compared with 2011.The individuals who were female, were younger, lived in urban, central or western regions, had medical insurance, had poor self-rated health and exhibited ADL impairment were more likely to use outpatient and inpatient services. The individuals who were female, were oldest-old, were rural residents, had poor self-rated health and exhibited ADL impairment relied more on private or primary medical institutions. In addition, transportation, medical expenses and the out-of-pocket ratio, as well as the urgency of the disease, were associated with the choice of providers. The universal medical insurance schemes improved health service utilization for the elderly population but had little impact on the choice of medical institutions.”

  1. The presentation of the writing seems unattending: the references are presented without uniformity, there are misspelled words, several undefined acronyms and the figures and tables appear as errors in the text.

Response:

Thank you for pointing out our mistake. We have improved the acronyms, the figures and other errors in the manuscript.

  1. It is presented as a limitation that a cross-sectional study was carried out instead of a longitudinal one, so that the associations found cannot be interpreted as causal associations. However, there were data from two other waves of the survey, why was a longitudinal design not used then?

Response:

Thank you for your kind reminds. Based on the Andersen’s behavioral model of health service use, we use the data of 2011, 2013 and 2015 to analyze the change trend of medical seeking behavior of the elderly in China, and analyze the influencing factors of medical seeking behavior of the elderly in 2015. However, we could not interpret the associations as causal associations. In future studies, we will use longitudinal data to make causal inferences about this issue.

Round 2

Reviewer 2 Report

The authors have responded adequately to most of the recommendations and it is now a manuscript that I can understand better. However, a couple of observations persist that are still not clear to me:
1.- The n of the data for the waves of 2011 and 2013 are still not presented, it is understood that the analysis of the factors was carried out on the data of 2015, but when presenting the trends in the use of health services in each wave I consider It is important to know if the n were comparable in all these years.
2.- The legend of table no.1 still does not clarify the associations presented, for example, 75.65% of the males used public outpatient services, the two asterics represent statistical significance when compared against primary outpatient or public inpatient? I guess the comparison is between public outpatient and public inpatient, but the problem is that it is not clear to the reader
3.- There are still problems in the references because in the text some include the initials of the authors, others include several authors and others do not, despite being the same number.